# Association between Bladder Outlet Obstruction and Bladder Cancer in Patients with Aging Male

**DOI:** 10.3390/jcm8101550

**Published:** 2019-09-27

**Authors:** Yu-Hsiang Lin, Chen-Pang Hou, Horng-Heng Juang, Phei-Lang Chang, Tien-Hsing Chen, Chien-Lun Chen, Ke-Hung Tsui

**Affiliations:** 1Department of Urology, Chang Gung Memorial Hospital at linkou, Taoyuan 333, Taiwan; linyh@cgmh.org.tw (Y.-H.L.); glucose1979@gmail.com (C.-P.H.); hhj143@mail.cgu.edu.tw (H.-H.J.); henryc@cgmh.org.tw (P.-L.C.);; 2School of Medicine, Chang Gung University, Taoyuan 333, Taiwan; skyheart0826@gmail.com; 3Graduate Institute of Clinical Medical Sciences, College of Medicine, Chang Gung University, Taoyuan 333, Taiwan; 4Department of Anatomy, School of Medicine, Chang Gung University, Kwei-shan, Tao-Yuan 333, Taoyuan, Taiwan; 5Department of Cardiology, Chang Gung Memorial Hospital at Keelung, Biostatiscal Consultation Center of Chang Gung Memorial hospital, Community Medicine Research Center, Keelung, Keelung 204, Taiwan

**Keywords:** benign prostate hyperplasia, bladder outlet obstruction, diabetes mellitus, prostatectomy, urinary tract infection, lower urinary tract symptoms

## Abstract

The associations between the treatment outcomes of benign prostatic hyperplasia/benign prostatic obstruction and lifelong health status, including urologic cancer incidence as well as geriatric adverse events (AEs), are unknown. This retrospective cohort study analyzed claims data collected during the period of 1997–2012 from Taiwan’s Longitudinal Health Insurance Database 2000. Patients who received transurethral resection of the prostate (TURP) were prioritized, and the remaining patients who were prescribed alpha-blockers were, subsequently, identified. Patients in the TURP and medication-only groups were further divided into two groups, according to the presence or absence of AEs during the first six-month follow-up. Outcomes of primary interest were all-cause mortality, occurrence of prostate cancer, transurethral resection of the bladder tumor, and radical cystectomy for bladder cancer. Compared with patients in the AE-free TURP group, those in the TURP with AEs had a higher risk of lifelong bladder cancer (subdistribution hazard ratio: 2.3, 95% confidence interval (CI): 1.56–3.39), whereas the risk of prostate cancer was comparable between the two groups (SHR: 1.2, 95% CI: 0.83–1.74). In the medication cohorts, patients undergoing alpha-blocker treatment who had AEs had a higher risk of all-cause mortality (hazard ratio: 1.63, 95% CI: 1.49–1.78) and a higher risk of lifelong bladder cancer (SHR: 2.72, 95% CI: 1.99–3.71) when compared with those without AE. Our study reveals that unfavorable treatment outcomes of benign prostate hyperplasia, whether caused by medication or surgical treatment, are associated with a higher incidence of bladder cancer. Unfavorable outcomes of surgical treatment are associated with higher risk of geriatric AEs, and unfavorable outcomes of medication treatment are associated with a higher risk of all-cause mortality.

## 1. Introduction

Benign prostatic hyperplasia (BPH), which is a key cause of lower urinary tract symptoms (LUTS) in older men, affects approximately 210 million men worldwide [1]. A study revealed that 50% of men developed pathological BPH at the age of 51–60 years [2]. Moreover, BPH/LUTS prevalence is expected to increase sharply in the coming decades [3]. The sequelae of BPH include decreased urinary flow and progression of voiding and storage symptoms, which eventually results in acute or chronic urinary retention (UR) [4]. BPH with moderate-to-severe LUTS considerably affects all aspects of the quality of life of men as they age [5]. A 50-year-old from the United States has an estimated risk of approximately 40% of undergoing therapeutic intervention (surgical or medical treatment) at some point during his lifetime [6]. Accordingly, billions of US dollars are spent annually to treat BPH/LUTS [7].

Both alpha-1 blockers and transurethral resection of the prostate (TURP) achieve favorable treatment outcomes in most patients with benign prostatic obstruction (BPO) [8]. Alpha-1 blockers are used for first-line treatment of BPO in men with LUTS [9]. In patients with unsatisfactory response to medication, TURP remains the dominant and definitive treatment for LUTS caused by BPH [10]. Some studies have also reported that TURP achieves favorable outcomes even for BPH patients with comorbidities such as UR, type 2 diabetes, and stroke [11,12,13]. Nevertheless, to the best of our knowledge, no studies have addressed the long-term sequelae for patients with “unfavorable” outcomes from either medical or surgical treatment. Therefore, we used Taiwan National Health Insurance Research Database (NHIRD) data to conduct a nationwide observational cohort study for investigating the correlation between the treatment outcomes of BPH/LUTS and lifelong health status, including urologic cancer incidence and geriatric adverse events.

## 2. Methods

### 2.1. Data Source

In this study, we used data from the Longitudinal Health Insurance Database 2000 (LHID2000), which is a subset of National Health Insurance research database (NHIRD.). LHID2000 contains the claims data of beneficiaries in the National Health Insurance (NHI) program of Taiwan, including details such as dates of inpatient and outpatient services, diagnoses, prescriptions, examinations, operations, and expenditures [14]. LHID2000 includes the claims data of 1,000,000 individuals randomly sampled from all NHI enrollees (a total of 23.75 million people) in 2000. The demographic characteristics (i.e., age and sex) between the populations derived from the NHIRD and LHID2000 are comparable. This study was approved by the Institutional Review Board of Chang Gung Memorial Hospital (Linkou Branch) (CMRP104-7810B).

### 2.2. Patient Identification and Definition of Exposure

We identified patients who had at least two outpatient diagnoses of BPH (International Classification of Diseases, Ninth Revision, Clinical Modification (ICD-9-CM) code: 600.xx) between 1 January, 1997 and 31 December, 2012. We first identified patients who received TURP between 1997 and 2012 by using the Taiwan NHI reimbursement codes of inpatient claims. The remaining patients who were prescribed alpha blockers for 90 days or longer within six months of the date of initial BPH diagnosis were classified as the medication-only group. For the TURP group, the index date was the discharge date following admission for TURP. For the medication-only group, the index date was the date of the initial prescription of alpha blockers for BPH.

Patients who met the following criteria were excluded: (1) age < 50 years, (2) a diagnosis of prostate cancer or bladder cancer before the index date, (3) episodes of prostate cancer, bladder cancer, or all-cause mortality within six months post treatment, and (4) follow-up of <6 months. Lastly, 6254 patients were included in the TURP group and 47,965 patients were included in the medication-only group.

The TURP and medication-only groups were further divided into two groups each, according to the presence or absence of adverse events (AEs) during the first six-month follow-up period. An AE was defined as follows: the occurrence of acute UR, urinary tract infection (UTI) with prescription of antibiotics, hematuria requiring endoscopic treatment, or bladder stone formation 31–180 days after the index date. Records of UTIs and bladder stone formation were extracted from the outpatient, emergency room, or inpatient claim data according to diagnostic codes (plus antibiotics) and Taiwan National Health Insurance (NHI) reimbursement codes, respectively. Urine retention (UR) was recorded only in outpatient and emergency room visits. The flow for patient selection is illustrated in Figure 1.

### 2.3. Covariates

The covariates were age, comorbidities, Charlson comorbidity index score [15], tissue ablation, number of outpatient visits in the previous year, admission and AEs in the previous three years, use of statin or nonsteroidal anti-inflammatory drugs in the previous year, and use of overactive bladder drugs or alpha blockers. A comorbidity was defined as two outpatient diagnoses or one inpatient diagnosis in the previous year. Many diagnoses of these diseases in the NHIRD have been validated in relevant studies [16]. The detailed International Classification of Disease, Ninth Modification (ICD-9-CM) diagnostic codes are listed in Table 1. Medications were identified from the claim data of outpatient visits or records of pharmacy refills.

### 2.4. Outcome Detection

Outcomes of primary interest were all-cause mortality, occurrence of prostate cancer, transurethral resection of the bladder tumor (TUR-BT), and radical cystectomy for treating bladder cancer. All-cause mortality was defined as withdrawal from the NHI program [17]. The occurrence of prostate cancer was verified by the approval to possess a catastrophic illness certificate. Definitions of prostate cancer have been widely reported in NHIRD studies [18]. TUR-BT and radical cystectomy operations were identified, according to the Taiwan NHI reimbursement codes of inpatient claim data. Secondary outcomes were long-term use of Benign Prostate Hypertrophy (BPH) medications (according to pharmacy refill records), recurrent adverse events (AEs)AEs, inguinal hernia, hemorrhoids, stroke, acute myocardial infarction, hip fracture, and spine fracture during the first three-year follow-up period. Each patient was followed from the index date to the date of event occurrence, date of death, or 31 December, 2012, whichever came first.

### 2.5. Statistics

To mitigate confounding factors due to treatment selection bias in this observational study, we adopted a propensity score matching method. The propensity score was the predicted probability of being in the AE group during the first 6-month follow-up, according to the values of covariates obtained using a logistic regression. Table 2 lists the variables selected to calculate the propensity score, where the follow-up year was replaced by the index date (Table 2). Each patient in the AE group was matched with two corresponding patients in the non-AE group. The matching was processed using a greedy nearest neighbor algorithm with a caliper that was 0.2-times the value of the standard deviation of the logit of the propensity score by applying a random matching order without replacement. The quality of matching was assessed using the absolute value of the standardized difference (STD) between the groups, where a value <0.1 was considered a negligible difference.

The difference in the risk of fatal outcomes (e.g., all-cause mortality) between the AE and non-AE groups was evaluated using the Cox proportional hazard model. The difference in incidence of nonfatal outcomes (e.g., Transurethral Resection of Bladder Tumor (TUR-BT)) between the groups was evaluated using the Fine and Gray sub-distribution hazard model, which considers all-cause mortality as a competing risk. The study group (AE compared with non-AE groups) was the only explanatory variable in the survival analyses. The within-pair clustering of outcomes after propensity score matching was accounted for by using a robust standard error, known as a marginal model [19].

A two-sided *p* of <0.05 was considered statistically significant, and no adjustment of multiple testing (multiplicity) was made in this study. All statistical analyses were performed using: Statistical Analysis System (SAS) (version 9.4, SAS Institute, Cary, NC, USA), including the procedures of “psmatch” for propensity score matching, “phreg” for survival analysis, and the macro “% cif” for generating a cumulative incidence function under the Fine and Gray sub-distribution hazard method.

## 3. Results

### 3.1. Study Population

The data of 66,782 patients from the NHIRD were analyzed, as shown in Figure 1. Overall, 6254 patients who underwent TURP and 47,965 patients who underwent alpha blocker treatment were eligible for analysis. Both of the cohorts were further sub-grouped into an AE group and a non-AE group. After matching, we divided the included patients into the following four groups for analysis: TURP with AEs (*n* = 1108), TURP without AEs (*n* = 2216), alpha blockers with AEs (*n* = 2633), and alpha blockers without AEs (*n* = 5266). Table 1 and Table 2 list the basic characteristics of the patients who underwent TURP and medication only, respectively. No significant difference (STD absolute value <0.1) was observed between the patients with AEs and those without AEs with respect to all of the covariates in either the TURP or medication-only groups.

### 3.2. Association between AEs and Treatment Outcomes

Table 3 details the relationship between the occurrence of post operation AEs and clinical outcomes during the follow-up period in the TURP group. Risk of all-cause mortality was comparable between the two groups (hazard ratio (HR): 1.07; 95% CI: 0.93–1.23), as shown in Figure 2A. However, patients who had an AE within six months following operation had a risk of bladder cancer formation requiring TUR-BT that was subsequently increased when compared with patients without AE (sub-distribution hazard ratio (SHR): 2.3; 95% CI: 1.56–3.39), as shown in Figure 2B. By contrast, the risk of prostate cancer formation remained similar between these groups (SHR: 1.20, 95% CI: 0.83–1.74). The risk of postop long-term medication dependence, both for alpha blockers and antimuscarinic medications, was also significantly different between the two groups, as indicated in Figure 2C. In addition, patients with post-operation AE had higher risk of long-term dependence on medications for LUTS, hemorrhoids, and hip fractures caused by accidental falls during the first three-year follow-up period.

Table 4 details the relationship between AEs following treatment with alpha blockers and clinical outcomes during the follow-up. Patients with AEs following alpha blocker treatment had a higher all-cause mortality rate (HR: 1.63, 95% CI: 1.49–1.78) compared with those without AEs, as indicated in Figure 3A. Patients with AEs following alpha blocker treatment also had a higher risk of bladder cancer formation requiring TUR-BT or radical cystectomy, as shown in Figure 3B,C. By contrast, the risk of prostate cancer formation remained similar (sub-distribution hazard ratio (SHR): 1.16, 95% CI: 0.84–1.61) between the two groups. In addition, patients with post-operation AEs had a higher risk of long-term dependence on medications for treating LUTS (Table 5).

## 4. Discussion

UR, Urinary tract infection (UTI), gross hematuria with bladder tamponade, and bladder stone formation are all possible sequelae of benign prostate obstruction (BPO) and are regarded as absolute indications of TURP [20]. However, it is frustrating for both physicians and patients if the previously mentioned complications reoccur despite treatment, regardless of whether the treatment is in the form of a surgical or medical intervention. According to the literature, unfavorable events of TURP include UTI (1.7–8.2%), UR (3–9%), hematuria with clot retention (2–5%), urethral strictures (2.2–9.8%), and bladder neck contractures (0.3–9.2%) [21]. The incidence of unfavorable events following treatment with alpha blockers varies among reports [22]. Nevertheless, no studies have focused on whether these unfavorable events affect older adults during long-term follow-up.

Before comparing the clinical outcomes of the four cohorts, 1:1 propensity score matching [23] was performed to ensure that the characteristics of both groups were similar and that more objective data could be obtained. Therefore, as detailed in Table 2 and Table 6, the distribution of parameters, including age, incidence of comorbidities, and Charlson comorbidity index, did not differ significantly between each two groups. The major finding of this research is that post-treatment AEs (whether following surgical or medical intervention) are significantly correlated with bladder cancer formation during lifelong follow-up. Bladder cancer is the ninth most common cancer worldwide, with an estimated 430,000 new cases in 2012 [24]. Bladder cancer resulted in 170,000 deaths globally in 2011, which demonstrates an increase from the 114,000 deaths in 1990 of 19.4% after adjusting for the increase in the total world population [25]. Data collected using the Taiwan Cancer Registry database suggest that the incidence of bladder cancer in the general population is approximately 0.21% [26]. However, the patients included in our study exhibited a higher bladder cancer incidence compared with the Taiwan general population. Patients with post-treatment AE even had a greater risk of bladder cancer than those without AE. This phenomenon indicates that an unfavorable treatment outcome for BPO is associated with an increased incidence of bladder cancer.

Multiple factors are associated with bladder cancer development. However, smoking tobacco is the main known contributor [27]. The number of cigarettes smoked, degree of inhalation, type of tobacco, use of filters, and smoking cessation all have significant relationships with bladder cancer development [28]. In addition, common candidate genes or pathways, such as carcinogenic metabolizing genes, DNA repair genes, apoptosis-related genes, and microRNA-related genes, have been studied widely and can contribute to the risk of bladder cancer [29]. Clinically, men with severe LUTS have a 64% higher relative risk of bladder cancer compared with men who report no LUTS. A recent study even revealed that patients with BPH who underwent TURP were at a higher risk of bladder cancer (HR = 6.17, 95%, CI = 3.68–10.3) than those who did not [30]. The sequelae of unfavorable BPO treatment outcomes, such as high post-voiding residual urine volume, repeat UTIs, chronic bladder inflammation, and chronic UR, all increase urothelial exposure to carcinogens and, thus, increase the risk of bladder cancer [31]. After chronic inflammation develops, it can mediate cancer pathogenesis by stimulating malignant cell growth, invasion, and metastasis through the recruitment of inflammatory cells and signaling molecules [32]. Therefore, an unfavorable treatment outcome for BPO is associated with an increasing bladder cancer incidence.

Similar to bladder cancer development, that of prostate cancer is positively associated with chronic tissue inflammation and bacterial infection [33]. Another study also reported that chronic tissue inflammation is associated with an increased risk of both prostate and bladder cancer, and a subgroup analysis by ethnicity suggested that the association was much stronger in Asian patients than in Caucasian patients [34]. A study pertaining to microbiomes indicated a prevalence of pro-inflammatory bacteria and uro-pathogens in the urinary tract of men with prostate cancer [35]. Urinary microbiomes influence chronic inflammation in the prostate and lead to prostate cancer development and progression [35]. Nevertheless, the findings of our study suggest that an unfavorable treatment outcome for BPO is not associated with prostate cancer development. Our explanation for this phenomenon is that prostate cancer development is mediated in part by chronic inflammation as well as by genetics and exposure to toxins in the environment. Thus, compared with unfavorable BPO treatment outcomes, other factors such as age, race, family medical history, lifestyle, and presence of metabolic syndrome might be stronger contributors to prostate cancer development.

Another notable finding of this study is that the treatment outcome of BPO had a substantial impact on the health status of male patients. In the medication-only group, patients with AEs, despite regular alpha blocker treatment, had a higher risk of all-cause mortality. By contrast, patients with AE following TURP also had increased risk of hemorrhoids, hip fracture caused by accidental falls, and medication dependence (alpha blockers or antimuscarinic medications) during a three-year follow-up. Many patients have persistent voiding dysfunction after surgical treatment for BPO. Older age, history of diabetes, history of cerebrovascular accidents, and preoperative antimuscarinic drug use are possible risk factors of continuing medical therapy following TURP [36]. We used 1:1 propensity score matching to eliminate the influence of the previously mentioned factors and concluded that the occurrence of AE within the six months following operation is also strongly associated with lifelong dependence on urologic medications.

This study has some limitations as a result of the NHIRD data structure. First, this database does not provide detailed personal information such as family cancer history, laboratory parameters, cigarette use, environmental toxin exposure, body mass index (BMI), and dietary habits, which are all confounding variables that influence prostate and bladder cancer development. Second, we used a strict criterion to divide our study population into two categories: patients with or without AEs. Thus, we could not assess how the duration, frequency, and severity of AEs affected treatment outcomes. Third, the use of laser prostatic vaporization or vapor-resection, which is not reimbursed by the Taiwan NHI, has only become increasingly common in the last decade [37]. Thus, patients receiving prostate laser treatment were not included in this database. However, despite these limitations, we believe that this is innovative and valid research. This study confirmed that unfavorable treatment outcomes of BPH increase the incidence of bladder cancer and have negative health effects among older men.

## 5. Conclusions

This nationwide database study reveals that unfavorable outcomes of BPH following either medication-based or surgical treatment are associated with a higher bladder cancer incidence. These outcomes, subsequently, lead to negative health effects in older men.

## Figures and Tables

**Figure 1 jcm-08-01550-f001:**
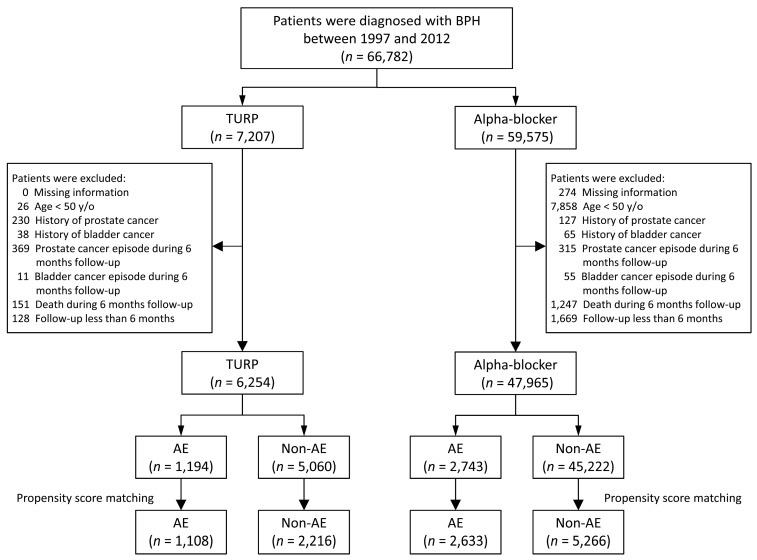
Patient selection.

**Figure 2 jcm-08-01550-f002:**
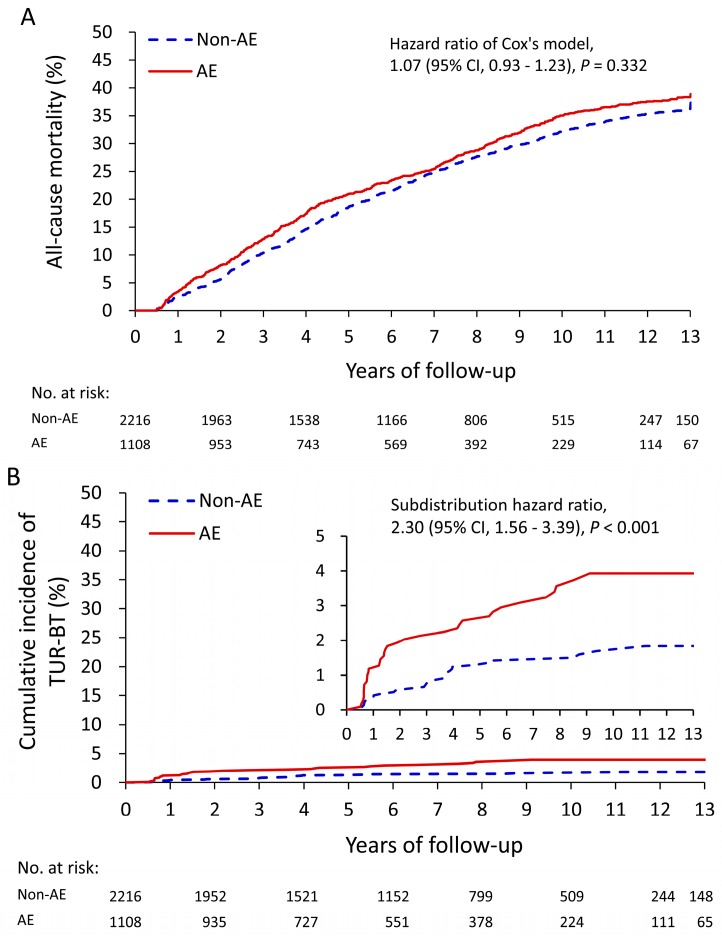
Unadjusted cumulative event rate of all-cause mortality (**A**), cumulative incidence of transurethral resection of the bladder tumor (**B**), and long-term use of medications for treating benign prostatic hyperplasia (**C**) in patients with and without adverse events during the first six-month follow-up in the TURP cohort.

**Figure 3 jcm-08-01550-f003:**
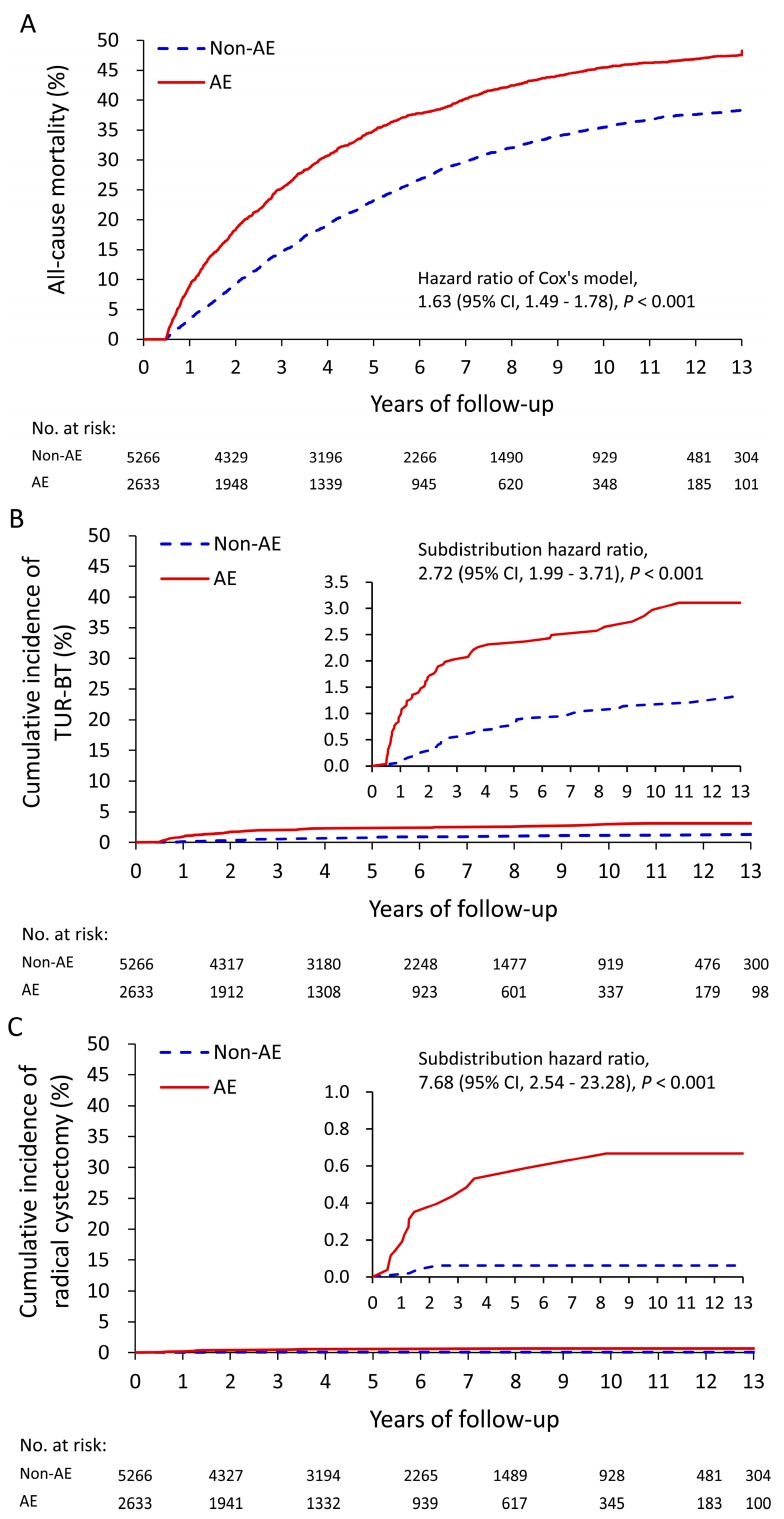
Unadjusted cumulative event rate of all-cause mortality (**A**), cumulative incidence of transurethral resection of bladder tumor (**B**), and cumulative incidence of radical cystectomy (**C**) in patients with and without adverse events during the first six-month follow-up in the medication-only cohort.

**Table 1 jcm-08-01550-t001:** ICD-9-CM codes used in the current study.

Variable	Code
Benign prostatic hyperplasia	600.xx, A360
Prostate cancer	185.xx (Catastrophic illness certificate)
Bladder cancer	188.9x (Catastrophic illness certificate)
Diabetes mellitus	250.xx
Hypertension	401.xx–405.xx
Hyperlipidemia	272.xx
Chronic obstructive pulmonary disease	491.xx, 492.xx, 496.xx
Parkinsonism	332.xx
Chronic kidney disease	580.xx–589.xx, 403.xx–404.xx, 016.0x, 095.4x, 236.9x, 250.4x, 274.1x, 442.1x, 447.3x, 440.1x, 572.4x, 642.1x, 646.2x, 753.1x, 283.11, 403.01, 404.02, 446.21
Ischemic heart disease	410.xx–414.xx
Stroke	430.xx–434.xx
Heart failure	428.xx
Alcoholism	V113, 291.xx, 305.0x, 357.5, 425.5, 303.xx, 571.0, 571.1, 571.2, 571.3, 980.0
Drug abuse	303.xx–305.xx
Urinary tract infection	599.0x, 595.0x
Hemorrhoids	455.xx
Acute myocardial infarction	410.xx
Hip fracture	820.xx
Spine fracture	805.xx, 806.xx

ICD-9-CM, International Classification of Diseases, Ninth Revision, Clinical Modification.

**Table 2 jcm-08-01550-t002:** Baseline characteristics of patients with BPH who received TURP grouped according to the presence or absence of AEs during the six-month follow-up period.

Variable	Before Matching			After Matching		
	AE (*n* = 1194)	Non-AE (*n* = 5060)	STD	AE (*n* = 1108)	Non-AE (*n* = 2216)	STD
Age (years)	72.9 ± 8.0	71.6 ± 8.0	0.160	72.7 ± 8.0	72.5 ± 7.8	0.021
Comorbidity						
Diabetes mellitus	227 (19.0)	867 (17.1)	0.049	203 (18.3)	413 (18.6)	−0.008
Hypertension	596 (49.9)	2106 (41.6)	0.167	534 (48.2)	1075 (48.5)	−0.006
Hyperlipidemia	142 (11.9)	450 (8.9)	0.098	124 (11.2)	244 (11.0)	0.006
Chronic obstructive pulmonary disease	229 (19.2)	657 (13.0)	0.169	191 (17.2)	361 (16.3)	0.025
Parkinsonism	34 (2.8)	93 (1.8)	0.067	24 (2.2)	51 (2.3)	−0.009
Chronic kidney disease	124 (10.4)	483 (9.5)	0.028	117 (10.6)	221 (10.0)	0.019
Ischemic heart disease	236 (19.8)	791 (15.6)	0.108	196 (17.7)	405 (18.3)	−0.015
Stroke	165 (13.8)	408 (8.1)	0.185	123 (11.1)	252 (11.4)	−0.009
Heart failure	54 (4.5)	166 (3.3)	0.064	41 (3.7)	92 (4.2)	−0.023
Alcoholism	6 (0.5)	10 (0.2)	0.052	1 (0.09)	5 (0.23)	−0.034
Drug abuse	3 (0.3)	13 (0.3)	−0.001	1 (0.09)	4 (0.18)	−0.025
CCI score	1.5 ± 1.7	1.2 ± 1.5	0.218	1.4 ± 1.6	1.4 ± 1.7	0.002
Tissue ablation						
5–15 g	850 (71.2)	3771 (74.5)	−0.075	791 (71.4)	1576 (71.1)	0.006
15–50 g	288 (24.1)	1086 (21.5)	0.063	269 (24.3)	538 (24.3)	0.000
>50 g	56 (4.7)	203 (4.0)	0.033	48 (4.3)	102 (4.6)	−0.013
Urologic event in the previous three years						
Urinary tract infection	484 (40.5)	1192 (23.6)	0.370	401 (36.2)	790 (35.6)	0.011
Urinary retention	495 (41.5)	1783 (35.2)	0.128	445 (40.2)	898 (40.5)	−0.007
Bladder stone	39 (3.3)	137 (2.7)	0.033	34 (3.1)	67 (3.0)	0.003
Urologic drug use in the previous three months						
Anti-muscarinic	114 (9.5)	488 (9.6)	−0.003	108 (9.7)	221 (10.0)	−0.008
Alpha-blockers	920 (77.1)	3817 (75.4)	0.038	851 (76.8)	1706 (77.0)	−0.004
Propensity score	0.235 ± 0.105	0.181 ± 0.085	0.569	0.216 ± 0.082	0.215 ± 0.081	0.008
Follow-up years	6.3 ± 3.8	7.3 ± 4.3	−0.225	6.5 ± 3.9	6.7 ± 3.9	−0.050

STD, standardized difference. BPH, benign prostatic hyperplasia. TURP, transurethral resection of the prostate. AE, adverse event. CCI, Charlson comorbidity index.

**Table 3 jcm-08-01550-t003:** Treatment outcomes during follow-up of patients with BPH who underwent TURP.

Variable	AE (*n* = 1108)	Non-AE (*n* = 2216)	AE vs. Non-AE	
			HR/SHR (95% CI)	*P*
**Primary Outcome at the End of Follow Up**				
All-cause mortality	431 (38.9)	827 (37.3)	1.07 (0.93, 1.23)	0.332
Prostate cancer	31 (2.8)	56 (2.5)	1.20 (0.83, 1.74)	0.337
TUR-BT	36 (3.2)	32 (1.4)	2.30 (1.56, 3.39)	<0.001
Radical cystectomy	5 (0.5)	1 (0.0)	NA	NA
**Secondary outcome during three-year follow-up**				
Medication dependence				
Anti-muscarinic	30 (2.7)	26 (1.2)	2.27 (1.48, 3.46)	<0.001
Alpha-blocker	112 (10.1)	172 (7.8)	1.31 (1.08, 1.59)	0.006
Inguinal hernia	40 (3.6)	59 (2.7)	1.35 (0.98, 1.87)	0.066
Hemorrhoids	132 (11.9)	195 (8.8)	1.39 (1.16, 1.66)	<0.001
Stroke	56 (5.1)	118 (5.3)	0.97 (0.74, 1.25)	0.788
AMI	8 (0.7)	19 (0.9)	0.80 (0.40, 1.57)	0.511
Hip fracture	21 (1.9)	20 (0.9)	2.26 (1.37, 3.71)	0.001

TURP, transurethral resection of the prostate. AE, adverse event. HR, hazard ratio. SHR, sub-distribution hazard ratio. CI, confidence interval. TUR-BT, transurethral resection of bladder tumor. NA, not applicable. AMI: Acute myocardial infarction.

**Table 4 jcm-08-01550-t004:** Treatment outcomes during follow-up of patients with BPH who received alpha blocker therapy.

Variable	AE (*n* = 2633)	Non-AE (*n* = 5266)	AE vs. Non-AE	
			HR/SHR (95% CI)	*p*
**Primary Outcome at the End of Follow Up**				
All-cause mortality	1272 (48.3)	2060 (39.1)	1.63 (1.49, 1.78)	<0.001
Prostate cancer	44 (1.7)	89 (1.7)	1.16 (0.84, 1.61)	0.354
TUR-BT	66 (2.5)	49 (0.9)	2.72 (1.99, 3.71)	<0.001
Radical cystectomy	15 (0.6)	3 (0.1)	7.68 (2.54, 23.28)	<0.001
**Secondary outcome during three-year follow-up**				
Inguinal hernia	55 (2.1)	96 (1.8)	1.12 (0.85, 1.47)	0.418
Hemorrhoids	202 (7.7)	403 (7.7)	1.04 (0.90, 1.19)	0.629
Stroke	143 (5.4)	262 (5.0)	1.06 (0.90, 1.26)	0.475
AMI	30 (1.1)	45 (0.9)	1.33 (0.91, 1.92)	0.137
Hip fracture	36 (1.4)	82 (1.6)	0.93 (0.67, 1.29)	0.651

AE, adverse event. HR, hazard ratio. SHR, sub-distribution hazard ratio. CI, confidence interval. TUR-BT, transurethral resection of the bladder tumor. AMI, acute myocardial infarction.

**Table 5 jcm-08-01550-t005:** Association between baseline characteristics and long-term use of medications for treating BPH among patients with BPH who received TURP (*n* = 6254).

Predictor	HR	95% CI	*p* Value
Age ≥ 75 years	1.29	1.07–1.56	0.008
BPH duration (years)	1.13	1.10–1.16	<0.001
Adverse event during the six-month follow up	1.45	1.18–1.79	<0.001
Hypertension	1.82	1.49–2.22	<0.001
Hyperlipidemia	1.51	1.17–1.94	0.002
Urinary tract infection in the previous three years	1.23	1.01–1.49	0.039
NSAIDs use in the previous year	1.35	1.11–1.63	0.002
Alpha-blocker use in the previous three months	1.61	1.21–2.14	0.001

BPH, benign prostatic hyperplasia. TURP, transurethral resection of the prostate. HR, hazard ratio. CI, confidence interval. NSAIDs, nonsteroidal anti-inflammatory drugs.

**Table 6 jcm-08-01550-t006:** Baseline characteristics of patients with BPH who received alpha blocker therapy grouped according to the presence or absence of AEs during the six-month follow-up period.

	Before Matching			After Matching		
Variable	AE (*n* = 2743)	Non-AE (*n* = 45,222)	STD	AE (*n* = 2633)	Non-AE (*n* = 5266)	STD
Age (years)	70.1 ± 10.3	66.7 ± 9.9	0.335	69.8 ± 10.3	70.1 ± 10.8	−0.023
Comorbidity						
Diabetes mellitus	641 (23.4)	7414 (16.4)	0.175	589 (22.4)	1193 (22.7)	−0.007
Hypertension	1264 (46.1)	17292 (38.2)	0.159	1192 (45.3)	2424 (46.0)	−0.015
Hyperlipidemia	309 (11.3)	5598 (12.4)	−0.035	300 (11.4)	604 (11.5)	−0.002
Chronic obstructive pulmonary disease	466 (17.0)	4752 (10.5)	0.189	412 (15.6)	853 (16.2)	−0.015
Parkinsonism	85 (3.1)	539 (1.2)	0.132	75 (2.8)	155 (2.9)	−0.006
Chronic kidney disease	324 (11.8)	3010 (6.7)	0.179	296 (11.2)	603 (11.5)	−0.007
Ischemic heart disease	484 (17.6)	5985 (13.2)	0.122	450 (17.1)	936 (17.8)	−0.018
Stroke	533 (19.4)	3580 (7.9)	0.340	460 (17.5)	934 (17.7)	−0.007
Heart failure	173 (6.3)	1067 (2.4)	0.195	143 (5.4)	312 (5.9)	−0.021
Alcoholism	33 (1.2)	347 (0.8)	0.044	32 (1.22)	74 (1.41)	−0.017
Drug abuse	19 (0.7)	238 (0.5)	0.021	19 (0.72)	40 (0.76)	−0.004
CCI score	1.6 ± 1.9	1.0 ± 1.5	0.385	1.5 ± 1.8	1.6 ± 1.9	−0.021
Urology event in the previous three years						
Urinary tract infection	893 (32.6)	4408 (9.7)	0.582	783 (29.7)	1541 (29.3)	0.010
Urinary retention	529 (19.3)	1693 (3.7)	0.502	419 (15.9)	780 (14.8)	0.031
Bladder stone	39 (1.4)	223 (0.5)	0.095	32 (1.2)	52 (1.0)	0.022
Urologic drug use in the previous three months						
Anti-muscarinic	79 (2.9)	675 (1.5)	0.095	74 (2.8)	162 (3.1)	−0.016
Alpha-blockers	—	—	—	—	—	—
Propensity score	0.121 ± 0.126	0.053 ± 0.053	0.705	0.105 ± 0.096	0.104 ± 0.095	0.003
Follow-up years	5.1 ± 3.8	6.7 ± 4.3	−0.404	5.1 ± 3.8	5.9 ± 3.9	−0.200

STD, standardized difference. AE, adverse event. CCI, Charlson comorbidity index.

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
