# Peer review of "Association between Bladder Outlet Obstruction and Bladder Cancer in Patients with Aging Male"

_jcm, 2019, doi:10.3390/jcm8101550_

Round 1

Reviewer 1 Report

The authors present this work about the association between bladder cancer and bladder obstruction. The manuscript is very well organize and clear, and the information presented in this work is of relevance for the scientific community. My only minor drawback is that a minor english review could improve the manuscript comprehension.

Author Response

Dear Editor :

Thanks to the editor for giving us the opportunity to make the article more intimate, the following changes are made according to the editors suggestions.

Reviewer 1

The authors present this work about the association between bladder cancer and bladder obstruction. The manuscript is very well organize and clear, and the information presented in this work is of relevance for the scientific community. My only minor drawback is that a minor english review could improve the manuscript comprehension.

Ans: We have followed the comment to English modified proof. Please see attach file 1.

Reviewer 2 Report

In the present study, the authors investigated the associations between the treatment outcomes of benign prostatic hyperplasia (BPH) and lifelong health status in patients who received transurethral resection of the prostate (TURP) and patients who were prescribed alpha-blockers. You concluded unfavorable treatment outcomes of BPH are associated with a higher incidence of bladder cancer, and unfavorable outcomes of medication treatment are associated with a higher risk of all-cause mortality. The study is interesting, but there is an illogical leap in what you said. There are some major points to propound:

You should discuss why unfavorable treatment outcomes of BPH are associated with a higher incidence of bladder cancer, additionally why unfavorable outcomes of medication treatment are associated with a higher risk of all-cause mortality.

Author Response

(The authors gave the same response as above.)
